# One Health, One Hive: A scoping review of honey bees, climate change, pollutants, and antimicrobial resistance

Etienne J. de Jongh[1,2,3,4,5], Sherilee L. Harper[2], Shelby S. Yamamoto[2], Carlee J. Wright[2], Craig W. Wilkinson[1], Soumyaditya Ghosh[3], Simon J. G. Otto[2,3,5,6]*

1 Faculty of Agriculture, Life, and Environmental Sciences, Department of Agricultural, Food and Nutritional Science, University of Alberta, Edmonton, Canada, 2 School of Public Health, University of Alberta, Edmonton, Canada, 3 HEAT-AMR (Human-Environment-Animal Transdisciplinary Antimicrobial Resistance) Research Group, School of Public Health, University of Alberta, Edmonton, Canada, 4 Faculty of Veterinary Medicine, University of Calgary, Calgary, Canada, 5 Antimicrobial Resistance–One Health Consortium, Calgary, Canada, 6 Healthy Environment Lead, Centre for Health Communities, School of Public Health, University of Alberta, Edmonton, Canada

* simon.otto@ualberta.ca

**Data Availability Statement:** All relevant data are within the manuscript and its Supporting Information files.

**Funding:** EJDJ: no number, University of Alberta Undergraduate Research Initiative, https://www.

## Abstract

Anthropogenic climate change and increasing antimicrobial resistance (AMR) together threaten the last 50 years of public health gains. Honey bees are a model One Health organism to investigate interactions between climate change and AMR. The objective of this scoping review was to examine the range, extent, and nature of published literature on the relationship between AMR and honey bees in the context of climate change and environmental pollutants. The review followed systematic search methods and reporting guidelines. A protocol was developed *a priori* in consultation with a research librarian. Resulting Boolean search strings were used to search Embase® via Ovid®, MEDLINE®, Scopus®, AGRICOLA™ and Web of Science™ databases. Two independent reviewers conducted two-stage screening on retrieved articles. To be included, the article had to examine honey bees, AMR, and either climate change or environmental pollution. Data, in accordance with Joanna Briggs Institute guidelines, were extracted from relevant articles and descriptively synthesized in tables, figures, and narrative form. A total of 22 articles met the inclusion criteria, with half of all articles being published in the last five years (n = 11/22). These articles predominantly investigated hive immunocompetence and multi-drug resistance transporter downregulation (n = 11/22), susceptibility to pests (n = 16/22), especially American foulbrood (n = 9/22), and hive product augmentation (n = 3/22). This review identified key themes and gaps in the literature, including the need for future interdisciplinary research to explore the link between AMR and environmental change evidence streams in honey bees. We identified three potential linkages between pollutive and climatic factors and risk of AMR. These interconnections reaffirm the necessity of a One Health framework to tackle global threats and investigate complex issues that extend beyond honey bee research into the public health sector. It is integral that we view these "wicked" problems through an interdisciplinary lens to explore long-term strategies for change.

ualberta.ca/current-students/undergraduate-research-initiative/index.html. The funders had no role in study design, data collection and analysis, decision to publish, or preparation of the manuscript.

**Competing interests:** The authors have declared that no competing interests exist.

## Introduction

The global rise of antimicrobial resistance (AMR) over the past 50 years presents troubling health projections for both public and environment sectors [1]. Antimicrobial resistance has global consequences for human health, resulting in approximately 700,000 deaths each year. By 2050, it is projected that the number of AMR-related deaths could rise to 10 million annually, with an estimated economic impact of $100 trillion USD [2]. Also at the forefront of global grand challenges lies climate change. The dire consequences of climate change have captured the focus and driven the collaboration of notable organizations such as NASA, the United Nations, and governments the world over [3–6].

Seeded into these critical contemporary issues are complex interactions that necessitate the conduct of interdisciplinary research [7,8]. Reports such as the World Health Organization (WHO) Antimicrobial Resistance Global Report, three recent Special Reports published by the Intergovernmental Panel on Climate Change (IPCC), and the Lancet Commission on Pollution and Health provide detailed insights into AMR, climate change, and environmental quality, respectively [1,9–12]. However, these reports neglect to substantially address these components through an interdisciplinary lens that links the three issues. Increasing communication between disciplines is not only helpful in understanding complex multidimensional problems, but is essential for implementing long-term solutions for mitigation [13,14].

While growing interest in areas such as One Health has helped bridge the topics of AMR, climate change, and environmental research, the majority of studies are still concerningly limited to the silo of each individual issue [1]. One Health is described as an approach to global health that focuses on linkages between the health of humans, animals, and the environment by improving intersectional communication and collaboration through research and policy [15].

Honey bees can serve as a model One Health organism to investigate the interactions between environmental change and AMR due to their inseparable symbiosis with the determinants of environmental health [16,17]. For example, environmental pollutants in water, soil, and air can negatively impact honey bee and hive health through leaching into pollen and honey foodstuffs [18,19]. Moreover, warming temperatures and other climatic factors related to climate change can increase the prevalence and spread of honey bee diseases and decrease the efficacy of antimicrobials in treating pests and pathogens [20–22]. Drug efficacy is further challenged by years of liberal antibiotic use [22,23], contributing to an increase in multidrug-resistant microorganisms. Apiaries globally are reporting greater colony losses than ever before [24,25]. It is generally believed that complex interactions between multiple environmental, pathogenic, and climatic factors are responsible for the majority of these losses, which have come to be referred to under the umbrella term of "colony collapse disorder" [26,27]. Interdisciplinary research into these interactions is therefore highly beneficial and inherently relevant to honey bee health.

How do environmental and climatic factors interact with each other to exacerbate AMR in honey bees? Given the limited evidence currently available, the objective of this scoping review was to examine the range, extent, and nature of published literature on the relationship between AMR and honey bees in the context of climate change and environmental pollutants through a One Health lens.

## Materials and methods

### Protocol and search strategy

The review followed systematic search methods outlined in the Joanna Briggs Institute (JBI) Reviewer's Manual and is reported according to the PRISMA Scoping Review reporting

guidelines [28–33]. A time-stamped protocol was developed *a priori* in consultation with a research librarian (S1 File). The PRISMA-ScR checklist is provided in S1 Checklist.

A comprehensive search strategy was developed to identify articles that discussed AMR in honey bees in the context of environmental or climatic factors. No search restrictions were placed on language, publishing date, or geography. An example search string for Embase® via Ovid® is shown in Table 1. The complete search strings (S1 Table) were used to search Embase® via Ovid®, MEDLINE®, Scopus®, AGRICOLA™ and Web of Science™ databases on July 10, 2019.

After downloading all retrieved articles within Mendeley© (Elsevier, Amsterdam, Netherlands), articles were collated and de-duplicated in DistillerSR® (Evidence Partners, Ottawa, ON, Canada) and screened for eligibility via a two-stage screening process by two independent reviewers. Article titles, abstracts, and key words were screened in the first stage, followed by full-text screening in the second stage. To be included, the article had to examine honey bees, AMR, and either climate change or environmental pollution (S2 File). Antimicrobial resistance

**Table 1. Search string used to search Embase® via Ovid® database for articles about honey bees, antimicrobial resistance, and environmental and/or climatic factors.**

| Component | Search Terms |
|---|---|
| **Honey Bees** | (bee **OR** bees **OR** honey?bee* **OR** honeybee* **OR** honey **OR** beekeep* **OR** apiar* **OR** arvicide* **OR** apis mellifera **OR** apidae **OR** (hive **AND** (health **OR** success **OR** collapse **OR** product* **OR** stability))) |
| **AND** | |
| **AMR** | (((resistan* **OR** stewardship) **AND** (antibiotic* **OR** antimicrobial* **OR** anti-microbial* **OR** anti-bacterial* **OR** antibacterial* **OR** anti?viral* **OR** antiviral* **OR** anti?fungal* **OR** antifungal* **OR** anti?helminthic* **OR** antihelminthic* **OR** anthelmintic* **OR** anti?parasitic* **OR** antiparasitic* **OR** parasiticide* **OR** biocid* **OR** antiseptic* **OR** disinfectant* **OR** sterilant* **OR** sterili?er* **OR** chemosterilant* **OR** multidrug **OR** multi?drug)) **OR** AMR **OR** XDR **OR** TDR **OR** super?bug* **OR** superbug*) |
| **AND** | |
| **Climatic Factors** | ((climat* **adj15** (chang* **OR** model?ing **OR** predict* **OR** resilience **OR** sensitivity)) **OR** (environment* **adj15** chang*) **OR** climate variability **OR** climatic variability **OR** global warm* **OR** greenhouse effect **OR** climate disaster **OR** (storm **NOT** (electrical **OR** autonomic **OR** thyroid*)) **OR** wind **OR** atmospheric pressure **OR** season* **OR** precipitation **OR** snow* **OR** ice **OR** humid* **OR** rain* **OR** flood **OR** drought **OR** wildfire* **OR** (heat **adj15** (wave* **OR** extreme* **OR** event)) **OR** temperature* **OR** cool **OR** cold **OR** weather **OR** ultraviolet radiation **OR** UV **OR** El Nino-Southern Oscillation **OR** El Nino **OR** La Nina) |
| **OR** | |
| **Environmental Factors** | (air pollut* **OR** persistent organic pollut* **OR** particulate matter **OR** atmospheric contamin* **OR** atmospheric pollut* **OR** volatile organic compound* **OR** volatile organic pollutant **OR** VOC **OR** VOCS **OR** ambient air pollution **OR** household air pollution **OR** criteria air pollutant* **OR** biological air pollutant* **OR** physical pollutant* **OR** chemical pollutant* **OR** gases **OR** ((fossil fuel **OR** arvicid*) **AND** pollut*) **OR** ((air **OR** water* **OR** soil) **AND** (contamin* **OR** toxic* **OR** environment* health **OR** quality **OR** disease* **OR** particulate* **OR** metal **OR** metals **OR** lead **OR** lead?II* **OR** Pb **OR** pb?+ **OR** zinc* **OR** Zn **OR** Zn?+ **OR** silver* **OR** Ag **OR** Ag+ **OR** copper* **OR** Cu **OR** Cu?+ **OR** Gallium* **OR** Ga **OR** Ga?+ **OR** cobalt* **OR** Co **OR** Co?+ **OR** Mercury* **OR** Hg **OR** Hg?+ **OR** Arsenic* **OR** As **OR** As?+ **OR** Nickel* **OR** Ni **OR** Ni?+ **OR** vehicle* **OR** automobile* **OR** exhaust **OR** motorway* **OR** roadway* **OR** highway* **OR** freeway* **OR** road* **OR** traffic **OR** urban **OR** Nox **OR** nitrogen oxides **OR** ozone **OR** particle*)) **OR** dust **OR** dusts **OR** PM?2?5 **OR** PM?10 **OR** ultrafine particle* **OR** polycyclic aromatic hydrocarbon* **OR** PAH **OR** POPS **OR** smog **OR** water pollut* **OR** (water* **AND** (potable **OR** healthy **OR** drink* **OR** safe **OR** suitab* **OR** palatable **OR** edible **OR** tap **OR** fresh **OR** supply **OR** microbial contamina*)) **OR** waterborne **OR** water?borne **OR** aquifer **OR** groundwater **OR** pesticid* **OR** herbicid* **OR** insecticid* **OR** acaricid* **OR** fungicid* **OR** molluscacid* **OR** larvicid* **OR** fumigant **OR** anti?fouling agent* **OR** agricultural chemical* **OR** agrochemical* **OR** (defoliant* **AND** (chemical* **OR** agent*)) **OR** (hazardous **AND** substance*) **OR** (toxic **AND** action*) **OR** chemically?induced disorder* **OR** furfural **OR** aculeximycin **OR** aluminum phosphide **OR** chromated copper arsenate **OR** CCA **OR** creosote) |

was defined as the ability of a pathogen to resist or reduce the effects of a drug or treatment meant to adversely affect its normal function [34]. Environmental change variables were defined as changes in climate due to natural or anthropogenic causes (climate change), or as an increase in organic or inorganic contaminants of soil, air, or water that alters their natural role or effect in honey bee colonies (environmental pollutants) [35]. Articles about season, weather, climate, and climate hazards in the context of climate change were also included. Honey bees were defined within the taxum *Apis mellifera* due to their agricultural importance, though articles using the terms "bees" or "honey bees" were considered relevant if no taxum was mentioned. The initial protocol required articles to include honey bees, AMR, climate change, and environmental pollutants. However, after screening articles to the data extraction level, a lack of articles containing all components prompted a revision of our inclusion criteria. This second round of screening included articles that studied honey bees, AMR and at least one of either climate change or environmental pollutants. This amendment was reflected within the protocol, which was re-time-stamped on December 9, 2019. The amendment was deemed necessary to provide sufficient evidence for discussion, to allow for better identification of gaps in literature, and to provide a more meaningful project outcome as a result. Articles were excluded if they were books, book chapters, theses, dissertations, or commentaries. Conflicts between reviewers were resolved via discussion if necessary.

### Data charting process and data items

Data regarding authorship, publication date, location of study, type of antimicrobial and target microbe, environmental and/or climatic factor assessed, research study design type, associated organizations, and outcomes of interest were extracted from relevant articles by two reviewers using DistillerSR®. Article information was exported to a pre-developed data extraction form within Excel® (Microsoft, Redmond, WA) for analysis (S2 Table). Articles were partitioned into thematic categories for further exploration, including: immunocompetence and multi-drug resistance (MDR) transporter downregulation, susceptibility to pests, and in-hive products.

Results were synthesized in tables, graphs, and narrative to present the comprehensive scope of current research in a concise and effective manner. Tables and figures present key findings in the results, while supplementary materials provide comprehensive results from the study to allow for replication in future research.

## Results

The initial search recovered 1,402 articles, with 1,146 remaining after deduplication (Fig 1). First-stage screening excluded 1,018 articles. 128 articles were eligible for second-stage, full-text screening, which reduced this number to 22. The majority of articles were excluded in this stage due to lacking mention of environmental variables or antibiotic resistance (n = 42), and failure to frame these topics in the context of honey bee health (n = 28). Despite our efforts to locate articles through both the University of Alberta and University of Guelph libraries, we were unable to locate full-text pdfs for 36 articles (S3 File). These articles were additionally requested through the University of Alberta and University of Guelph interlibrary loan systems to ensure minimal loss of articles. This process returned six additional articles that were screened, but 36 could not be obtained and were excluded.

### Characteristics of sources of evidence

Twenty-two articles met the inclusion criteria and were included in our analysis. An overview of these articles is included in Table 2, while a complete listing of included articles and study characteristics is available in S2 Table. Articles were published between 1993 and 2019. Research on

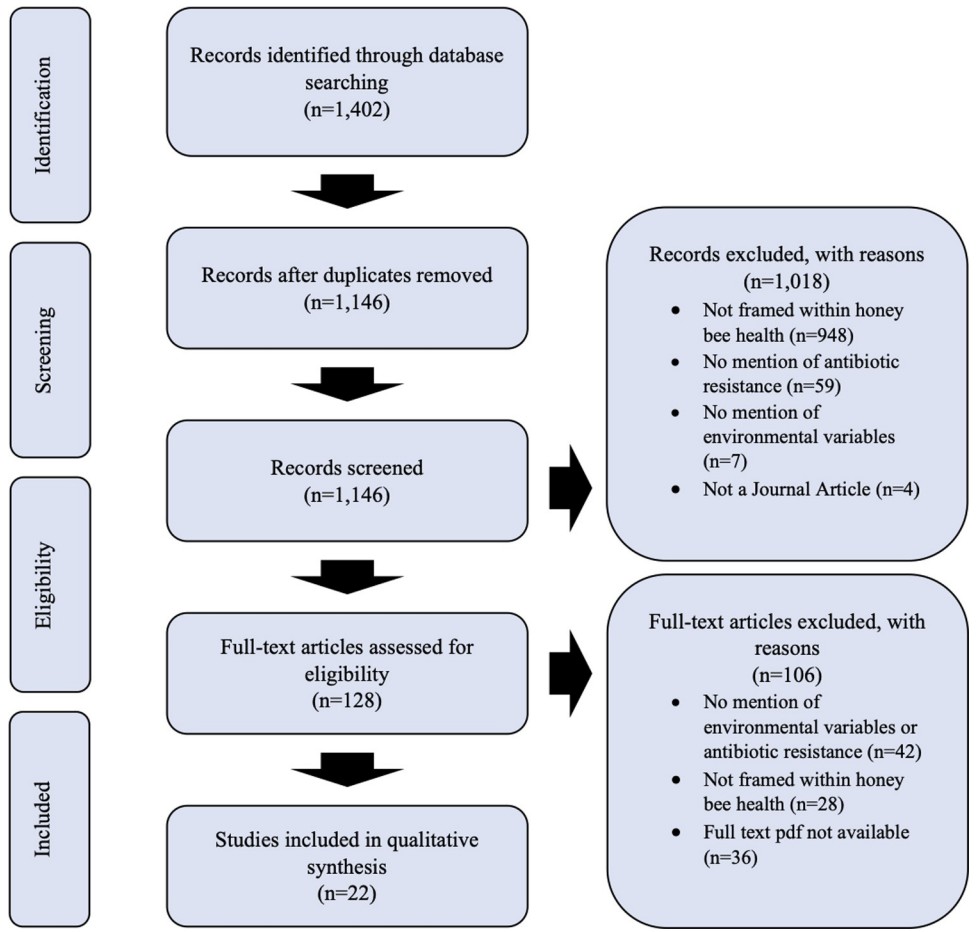

**Fig 1. PRISMA-ScR flow diagram of study selection process for the systematic scoping review of the impacts of climate change, environmental pollution, and antimicrobial resistance on honey bee health.**

AMR and effects of environmental change in honey bees steadily increased in recent years with half (n = 11/22) of included articles published in the last five years alone (2014–2019) (Fig 2).

Fig 3 shows the study location in a global context. Article publication represented research from ten countries that was distributed globally. While some articles did not specify a geographical origin (n = 4), the majority of publications occurred in high-income nations (n = 13; Czech Republic, Germany, Italy, Japan, Norway, Spain, United States) [56]. The United States constituted the largest proportion of location-specific publications (n = 6). A large proportion of articles also came from Europe, with a total of seven articles spread over six European countries (Germany, n = 2; Czech Republic, n = 1; Italy, n = 1; Norway, n = 1; Spain, n = 1; Turkey, n = 1.

Out of the 22 articles, 64% (n = 14/22) followed an experimental study design, with the rest being observational or descriptive studies (n = 16), or review articles (n = 2). There were relatively few studies with broader scope that investigated AMR and environmental change from a global or ecological perspective.

## Synthesis of results

Table 3 summarizes environmental factors of interest by climatic or pollutive basis. Environmental factors of interest varied greatly, with environmental insecticides being the most common pollutive factors (n = 7) and indirect geographical differences (different climate zones as

**Table 2. Summary of article characteristics captured by this study and deemed eligible for review.**

| Reference number | Author(s) | Year | Location | Hive/honey bee health aspects of concern | Antimicrobial | Target microbe | Climate variable of interest | Environmental quality factor of interest |
|---|---|---|---|---|---|---|---|---|
| [20] | Regueira Neto et al. | 2017 | Tamandare, Brazil | Immunocompet-ence, self-treatment | Red propolis, gentamicin, imipenem | *Escherichia coli, Pseudomonas aeruginosa, Staphylococcus aureus* | Precipitation, Seasonal variability (Wet vs Dry season) | -- |
| [36] | Ueno et al. | 2018 | 17 prefectures in Japan | American foulbrood | Mirosamicin, Oxytetracycline, Tylosin, Lincomycin, chloramphenicol, streptomycin, erythromycin | *Paenibacillus* larvae | Geography/general variations in climate | -- |
| [37] | Ebrahimi, and Lotfalian | 2005 | Shahrekord, Central Iran | Honey bee dysbiosis | Gentamicin, Streptomycin, Kanamycin, Amikacin Penicillin Chloramphenicol, Nalidixic Acid, Oxytetracycline, Erythromycin, Vancomycin, and Nitrofurantoin | *Escherichia coli, Staphylococcus aureus* | Temperature, seasonal variability (Spring vs Fall) | -- |
| [38] | James and Xu | 2011 | Not Stated/ Global | Immunocompetence, antimicrobial peptides, behavioural immunity | Antimicrobial peptides, reactive oxygen species, RNA interference | Bacteria, viruses, fungi, parasites | -- | Environmental pesticides, botanical insecticides (Acacia senega extract/ Artemisia annua extract/ Azadirachtin/Quercetin/Terpinen-4-ol), inorganic insecticides (Sodium tetraborate), insect growth regulators (Buprofezin/Fenoxycarb/ Flufenoxuron/Pyriproxyfen), neonicotinoids (Imidacloprid), organochlorines (Endosulfan/ Dieldrin), organophosphates (Dimethoate/Malathion/Quinalphos) |
| [39] | Travis et al. | 2014 | Not Stated/ Global | General honey bee morbidity related to increasing agriculture, such as pesticide use and monoculture | General/not stated | General/not stated | -- | General Insecticide and pesticide use associated with intensive agriculture |
| [40] | Bernal et al. | 2011 | Marchamalo, Spain | American foulbrood | Tylosins A, B, C, D | *Paenibacillus* larvae | Temperature, light | -- |
| [41] | Hawthorne et al. | 2011 | United States | American foulbrood, Varroa mite, Multidrug resistance transporters | Coumaphos, t-fluvalinate, oxytetracycline | *Paenibacillus* larvae, varroa mite | -- | Environmental insecticides, neonicotinoids (imidacloprid, acetamiprid, and thiacloprid) |
| [42] | Guseman et al. | 2016 | United States | Nosema, Multidrug resistance transporters | Verapamil, pristine, fumagillin, quercetin | *Nosema* sp. | -- | Environmental ivermectin and ivermectin-like pesticides, neonicotinoids |
| [43] | Brandt et al. | 2016 | Germany | Immunocompet-ence | Honey bee hemolymph | General/not stated | -- | Environmental neonicotinoids (thiacloprid, imidacloprid, and clothianidin) |
| [44] | Brandt et al. | 2017 | Germany | Immunocompet-ence | Honey bee hemolymph | General | -- | Environmental neonicotinoids |
| [45] | O'Neal et al. | 2019 | United States | Immunocompet-ence, social immunity | Innate antimicrobials | Viruses | -- | Environmental fungicides (chlorothalonil) |
| [46] | Prodelalová et al. | 2017 | Czech Republic | General viral infection | Peracetic acid, iodophors | *Paenibacillus* larvae, deformed wing virus, Sacbrood virus, and slow bee paralysis virus, black queen cell virus, acute paralysis complex viruses | Temperature | -- |
| [47] | Ozkirim, Aktas, and Keskin | 2007 | Turkey | American foulbrood | Sulbactam ampicillin, amoxicillin clavulanic acid, tobramycin, erythromycin, azithromycin, and rifampin | *Paenibacillus* Larvae | Geography/general variations in climate | -- |

*(Continued)*

**Table 2.** (Continued)

| Reference number | Author(s) | Year | Location | Hive/honey bee health aspects of concern | Antimicrobial | Target microbe | Climate variable of interest | Environmental quality factor of interest |
|---|---|---|---|---|---|---|---|---|
| [48] | Alippi et al; | 2005 | Not Stated/Global | American foulbrood | Tylosin | *Paenibacillus larvae, Pseudomonas aeruginosa, Escherichia coli, Staphylococcus aureus* | Geography/general variations in climate | -- |
| [49] | Erler and Moritz | 2015 | Not Stated/Global | American foulbrood, European foulbrood, varroa mite, deformed wing virus immunocompet-ence, chalkbrood, self-medication. | Beeswax, bee food jelly including royal jelly, bee venom, resin, propolis | *Enterococcus faecalis, Paenibacillus larvae*, acute bee paralysis virus, black queen cell virus, deformed wing virus, sacbrood virus, *Paenibacillus alvei, Galleria mellonella, Apis flavus, Aspergillus fumigatus, Aspergillus niger, Nosema apis, Nosema ceranae, Aethina tumida, Oecophylla smaragdina*, insects, dead mammals | Temperature, precipitation, climate type | -- |
| [50] | Chaimanee et al. | 2013 | Thailand | Nosema | Immunocompetence, Antimicrobial peptides | *Nosema ceranae* from Canada and Thailand | Geography/general variations in climate | -- |
| [51] | Bastos et al. | 2007 | Brazil | American foulbrood | Propolis, Vancomycin, Tetracycline, Tylosin | *Paenibacillus larvae* | Indirect, general climate affecting hive product antimicrobial strength | -- |
| [52] | Krongdang et al. | 2017 | United States | American foulbrood | Oxytetracycline, tetracycline, tylosin, lincomycin | *Paenibacillus larvae* | Geography/general variations in climate | -- |
| [53] | Gregorc et al. | 2012 | United States | Varroa mite, immunocompet-ence | Antimicrobial peptides (abaecin, hymenoptaecin, defensin1) | Deformed Wing Virus | | Environmental pesticides (chlorpyrifos, imidacloprid, amitraz, fluvalinate, coumaphos, myclobutanil, chlorothalonil, glyphosate, simazine) |
| [23] | Tian et al. | 2012 | United States | American foulbrood, European foulbrood, gut dysbiosis | Oxytetracycline | *Melissococcus pluton, Paenibacillus larvae* | Geography/general variations in climate | Environmental broad-spectrum antimicrobial exposure |
| [54] | Loglio | 1993 | Italy | Varroa mite | Fluvalinate | Varroa mite | Temperature, seasonal variability, sunlight, altitude, climate type | -- |
| [55] | Dickel et al. | 2018 | Norway | Immunocompetence | Thiacloprid | *Enterococcus faecalis* | -- | Environmental neonicotinoid thiacloprid |

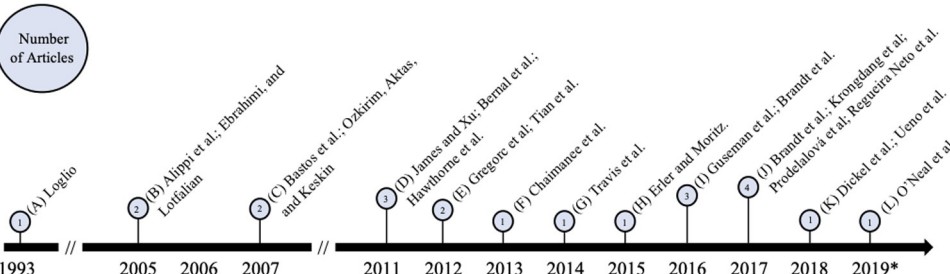

**Fig 2. Timeline of study publication dates for articles on honey bee health, antimicrobial resistance, climate change, and environmental pollution.** Articles are organized by year of publication and represented in quantity by the length of the pin above each respective year. The number of articles per year is included inside each pinhead. *Note 2019 was an incomplete year because the article search was conducted in July 2019.

a result of different geographical locations) accounting for the majority of climatic factors (n = 6). Although most articles revealed potential indirect links to AMR in honey bees, few articles directly linked specific pollutive variables to AMR, the most common of which was the effect of neonicotinoids (n = 6).

The 22 articles can be broadly divided into three thematic categories based on the focus of the study and linkage of AMR to environmental factors: 1) immunocompetence and MDR transporter downregulation; 2) interactions with pest susceptibility; and 3) influences on in-hive antimicrobial properties (categorization shown in Table 4).

**Immunocompetence and MDR transporter downregulation.** Of these 22 articles, nine focused on immunocompetence [20,38,43–45,49,50,53] and two investigated the downregulation of MDR transporters [41,42]. Combined, these eleven articles studied the synergistic effects of pesticides and climatic factors on honey bee innate immunity inhibition. Most

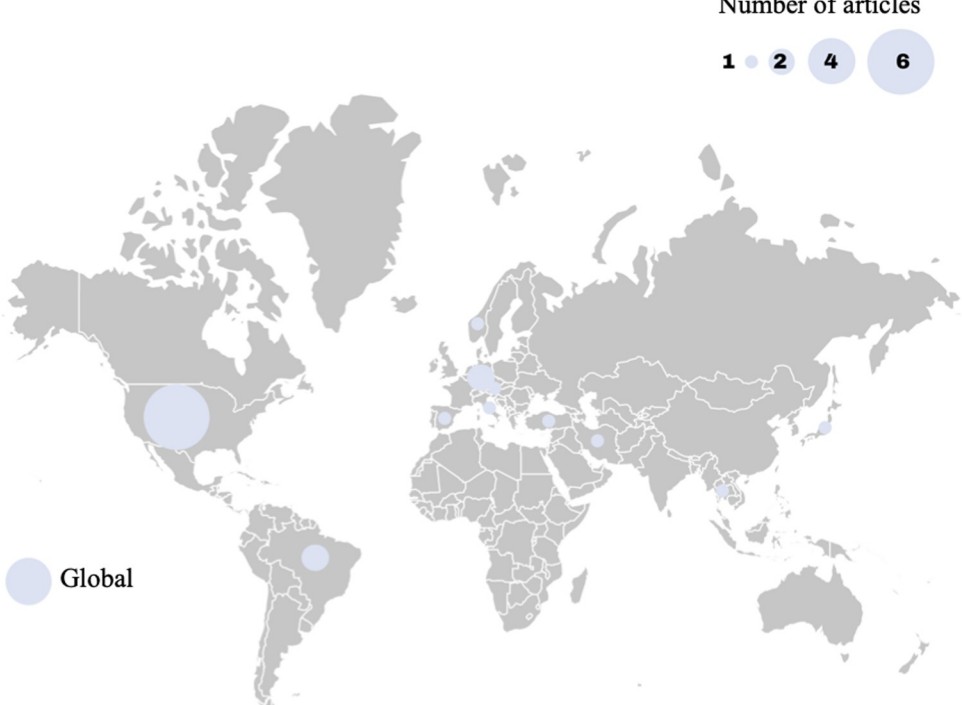

**Fig 3. The global distribution of the study locations included in the review presented as the number of studies by country with article numbers represented by the relative circle size.**

**Table 3. Summary of environmental factors of change in the included articles.**

| Article Environmental Factor of Change | # of relevant articles | Article reference |
|---|---|---|
| **Climatic Factors** | | |
| Season | 3 | [20,37,54] |
| Geography | 6 | [23,36,47,48,50,52] |
| Temperature | 5 | [37,40,46,49,54] |
| Sunlight | 2 | [40,54] |
| Precipitation | 2 | [20,49,49] |
| General/Climate type | 3 | [49,51,54] |
| **Pollutive Factors** | | |
| Pesticides | 4 | [38,39,42,53] |
| Insecticides | 7 | [38,39,41–44,55] |
| Fungicides | 1 | [45] |
| Other/General | 2 | [23,39] |

articles found correlations between exposure to antibiotics or pathogens and decreasing honey bee immune function. One article found an increase in immune function when exposed to contaminants and infection, and one final article noted that dual exposure of pathogens and pesticides may increase transmission of disease [38,55]. Most articles focused on alterations in honey bee immunocompetence resulting from the inhibition of immune-essential endogenous microbiota within the gastrointestinal tract [20,38,43–45,49,50,53]. These articles described defensive reactions on the part of the biota (e.g. drug efflux, gene expression) to pollutants and environmental contaminants, as well as inhibition of these defensive mechanisms. Several articles explored alteration of MDR transporters, which are natural efflux pumps present in the

**Table 4. Summary of article characteristics by thematic category.**

| Article General Topic of Interest | | # of relevant articles | Article reference |
|---|---|---|---|
| **Immunocompetence and multidrug resistance (MDR) transporter downregulation** | | | |
| Immunocompetence | | 9 | [20,38,43–45,49,50,53,55] |
| MDR transporter downregulation | | 2 | [41,42] |
| Increased morbidity | | 10 | [20,38,41–45,49,50,53] |
| Increased Transmission | | 1 | [38] |
| **Susceptibility to pests** | | | |
| Parasites | Varroa Mite | 4 | [41,49,53,54] |
| Fungi | *Nosema* | 2 | [42,50] |
| | Chalkbrood (*Ascosphaera apis*) | 1 | [49] |
| | General | 1 | [38] |
| Bacteria | American foulbrood (*Paenibacillus larvae*) | 9 | [23,36,40,41,47–49,51,52] |
| | European foulbrood (*Melissococcus plutonius*) | 2 | [23,49] |
| | Other | 1 | [55] |
| Viruses | | 3 | [38,45,53] |
| **Hive Products** | | | |
| | Brazilian Red Propolis | 3 | [20,49,51] |
| | Other/General | 1 | [49] |

cells of almost all animal species [41,42]. They pump many different classes of harmful compounds out of the cell, such as heavy metals, pesticides, and in some cases, antimicrobials [57]. Exposure to one of these compounds can trigger an upregulation of MDR efflux pump expression, thereby increasing resistance to multiple other types of compounds without direct exposure. In this way, MDR transporters can have substantial impact of the efficacy of drug dosages [42]. No article extrapolated this effect to the development of AMR.

**Susceptibility to pests.** Most studies investigated bacterial infections, with almost half of all articles focusing on *Paenibacillus larvae*, the causative agent of American foulbrood (n = 9/22) [23,36,40,41,47–49,51,52]. *Melissococcus plutonius*, the causative agent of European foulbrood, and *Enterococcus faecalis* was also studied [23,49,55]. The parasitic mite *Varroa destructor* (n = 4/22) [41,49,53,54] and the fungal genus *Nosema* (n = 2/22) [42,50] received some marginal exploration. These articles linked increased pollutants to reduced honey bee health in the form of antimicrobial peptide (AMP) expression modulation. Antimicrobial peptides are critical to insect immune defence, and by altering their transcription or expression, environmental pollutants may lead to increased infection and transmission of pests and pathogens [38]. Articles largely neglected to evaluate how this increase in disease may necessitate the need for increased drug treatment in the hive and to the development of AMR. Articles that predominantly focused on *V. destructor* infection investigated also investigated morbidity as a result of deformed wing virus infection due to the strong association between these two pathogens [58]. Morbidity as a result of Varroa mite infection often occurs due to secondary infection via deformed wing virus, *Escherichia coli*, or other bacterial or viral infections [58]. Therefore, most papers included in this review investigating pest susceptibility explored more than one pathogen at a time. The strong association between pest exposure and immune response, combined with the two-punch approach of most honey bee parasites (destruction of the cuticle followed by secondary viral or bacterial infection), and the broad-spectrum nature of honey bee immune factors resulted in significant overlap between articles binned under pest susceptibility and immunocompetence.

**In-hive products.** The third thematic category explored by this study was the self-administration of in-hive antimicrobial products on AMR. Three articles were included on this topic, all of which discussed the effect of the hive product propolis, an antibiotic and sealant made by the honey bees from resinous plant products, beeswax, and salivary enzymes [20,49,51]. Two of these three articles focused exclusively on the use of proplis [20,51], while one also investigated all-natural, pharmaceutically active compounds made and used by honey bees in the hive [49]. In regards to climatic variables, one article investigated seasonality and another investigated geographical origin as factors that impact the efficacy of propolis [20,51]. Together, these found that propolis was more inhibitory to bacteria, particularly *P. larvae*, when it was sourced from Brazil during the dry season. The remaining article looked how environmental factors influence self medicative behaviour among honey bees [49].

## Discussion

This study synthesized current interdisciplinary research on AMR, climate change, and environmental pollution in honey bees through a One Health lens in order to characterize past studies and identify potential avenues for future research. The scoping review identified 22 articles published between 1993 and 2019 that examined how interactions between climatic, pollutive, and microbial factors influenced honey bee health through AMR risk and development. Most of these studies were experimental, indicating that research in this area is largely empirical and topically isolated. In general, articles described linkages between environmental factors such as temperature or insecticide pollution and the ability of honey bees to resist or

treat hive infection, either at the colony or individual bee level, or at the biological or behavioural level. However, broad research on the linkage between AMR, climate change, and environmental pollutants on honey bee health was generally lacking, indicating a future need for interdisciplinary research in this field.

Honey bee immunity is complex and dependent on both behavioural and biological factors outside of, and within, the honey bee. Our study identified an opportunity for further investigation of immunocompetence and MDR transporter regulation as a consequence of environmental determinants. The relationship between immune function and MDR transporter regulation is pertinent to the field of AMR for a number of potential reasons. Firstly, any resistance acquired by honey bee cells via MDR transport upregulation could possibly increase the risk of AMR in symbiotic microbes [59,60]. Bacterial pathogens can acquire resistance genes through horizontal genetic transfer (HGT) [60]. There is evidence that insects transfer genetic material bidirectionally through HGT with intracellular primary endosymbiont bacteria within polyploid bacteriocyte cells [61]. Evidence of exchange of bacterial genes with fungal pathogens by HGT further strengthens this possibility [62], but specific evidence of the transfer of AMR genes through these mechanisms remains largely unstudied. As this theme did not emerge from the papers included in our scoping review, evaluation of its possibility for honey bees is outside the scope of this paper, but presents an intriguing area of interest for future One Health research.

Secondly, honey bee cell membrane transporters may reduce microbial exposure to administered antimicrobials. Natural honey bee cell membrane transporters remove intracellular compounds from the cytoplasm [57]. When pesticides are introduced to the hive, these transporters are activated to prevent the compounds from accumulating. Both pesticides and antimicrobials (including vital acaricides such as coumaphos) are substrates of these transporters [41,42]. As a result, pesticide-induced upregulation of these transporters may concurrently accelerate the removal of antimicrobials from the cell and decrease the intracellular concentration. With less antimicrobials circulating within the honey bee cells, intracellular pathogens such as *Nosema* spp. and pathogens that live within the body cavity such as *Ascosphaera apis* may be exposed to lower dosages during this upregulation of membrane transporters [61,62]. By "shading" potential pathogens from antimicrobial treatment, there presents an increased risk for AMR development by the microbes. A similar effect has been studied in the public health sector through the use of small colony variants of *Staphylococcus aureus*, whereby the microbe is theorized to shelter from antimicrobial treatment within host cells to increase resistance against treatment and allow recurring infections [63,64]. One article in our study highlighted the synergistic effect of simultaneous exposure to contaminants and pathogens [55]. Although this article demonstrates linked immune responses between two distinct etiological agents, the specific pathway was not explored and represents an opportunity for future study [55].

Lastly, with a decrease in honey bee immunity, pathogens are able to more quickly spread and develop inside the hive. Articles within our study primarily focused on immunity as a factor of honey bee endogenous microbiota, highlighting correlations between environmental pollutants and changes in microbiota function. These microbiota have been found to be exceptionally important both in honey bee pathogenic defence, as well as in recovery [65]. Small changes in the immune function of the honey bee linked to changes in these microbes can have drastic effects on the ability of honey bees to fight off disease. However, the articles in this study failed to evaluate how an adjustment in immunity may correspond to an increased risk of AMR. Notably, human studies have shown that a compromised immune system increases the risk of AMR emergence [66,67]. This can be due to inhibition of synergistic actions between the immune system and the antimicrobial in reaching an effective minimum

inhibitory concentration at the site of infection, an overall increase in disease prevalence, or a higher rate of mutation resulting from unhindered population growth. However, these connections are absent in the articles in this study, and therefore there remains the opportunity to address these connections in the future.

Our scoping review exposed correlations between environmental factors and an increased susceptibility of honey bees to disease. The predominant cause of vulnerability in the hive was due to modulation of AMPs by environmental pollutants. These peptides serve a critical role in innate defences against pathogens in all insects, including honey bees [68]. The effect of AMP on bacteria and viruses was a key focus of included articles due to the high incidence of American foulbrood (a bacterial infection) and Varroa Mite, which normally increase morbidity in the hive through secondary bacterial and viral infections [53]. Therefore, because most articles investigated morbidity as a result of bacteria and viruses either directly or indirectly, it follows that AMPs, the primary defence against these organisms, would also be investigated. As shown in human and livestock animal studies, an increase in disease susceptibility inevitably corresponds to an increase in antimicrobial drug treatment, with a subsequent increased risk of AMR [69–71]. Although increased antimicrobial usage is commonly inferred to correlate with an increased risk of AMR, none of the studies in this review investigated this connection. Therefore, there remains an opportunity to holistically connect evidence streams between disease susceptibility, treatment requirement, and risk of AMR to determine their interdependencies.

Although external antimicrobial treatment by beekeepers was the primary focus of research included in this review, our study revealed an increased interest in zoopharmacognostic (self-medicating) behaviours within the hive itself. While normal drug treatment in apiaries occurs once or twice per year in the spring and fall, self-medication processes by honey bees themselves within the hive are continuously implemented [72]. Additionally, honey bee self-medication utilizes products within the hive that are prone to variable strength and efficacy, partly due to outside factors. Our study exposed some contributors to this antimicrobial variance, namely temperature and seasonality. However, domestication has led to some additional challenges and considerations, such as the mixing of honey bees and antimicrobial products (e.g., honey and propolis) from multiple geographic sources. Given the sensitivity of hive products to climatic conditions, the relocation of honey bees to new climates and environments may alter the antimicrobial properties and efficacy of hive products. There is an opportunity to investigate how the alteration of these products may influence the ability of colonies to appropriately self-medicate. Despite this growing concern, we did not identify any studies that directly correlated honey bee hive product self-medication with an increased threat of AMR. Given that inconsistent antimicrobial strength can lead to AMR, and environmental conditions have been shown to contribute to antimicrobial inconsistency both in bees as well as the general population [20,73], connecting these two areas remains an opportunity for future interdisciplinary research.

## Strengths and limitations

While all literature reviews face the possibility of failing to capture all eligible articles, we aimed to minimize this risk by following a rigorous, systematic approach [74]. We adopted a search strategy without language limitations in order to reflect the global breadth of the issues at hand. However, this global undertaking resulted in the necessary exclusion of 36 articles that were deemed eligible through abstract screening but were not available to us for full-text review (S3 File). We recognize that 8/22 included articles were observational/descriptive studies or review articles, and less useful than the 14 experimental studies for identifying causal relationships. We also recognize one article with a questionable link between AMR and climate change or environmental pollution. The Prodelalová et al. (2017) paper used a surrogate virus

to assess the effectiveness of disinfectants against the viruses of interest (picornaviruses) at different temperatures. The experimental model itself was tenuous and did not factor largely into our findings. However, the novel insights derived from this study allowed for the identification of multiple literature gaps and future areas of interdisciplinary research and still illustrate the usefulness of honey bees as an organism to determine the One Health impacts of AMR, climate change, and environmental pollution.

## Conclusions

This study mapped current literature investigating the relationship between AMR and honey bees in the context of climate change and environmental pollutants through a One Health lens. We identified considerable potential for further interdisciplinary research to holistically correlate environmental influences on honey bee immunity, disease susceptibility, and self medicative behaviours on AMR risk. Despite the immense agricultural and economic significance of honey bees globally, we identified a lack of literature on honey bee health in the context of AMR. Our findings provide the basis for future research to understand the complex linkages of AMR, climate change, environmental pollution and honey bee health in the context of One Health. This study will contribute to the growing body of One Health and interdisciplinary research to find novel solutions for global "wicked" problems beyond the beehive.

## Supporting information

**S1 Checklist. Completed checklist.**
(PDF)

**S1 Table. Screening questions that define the inclusion and exclusion criteria used in the two-level screening process by two independent reviewers.**
(PDF)

**S2 Table. Data extraction table of complete study characteristics of included aritlces.**
(XLSX)

**S1 File. Protocol outlining the systematic scoping review created using JBI guidelines and following the PRISMA-ScR checklist–time-stamped on December 19, 2019.**
(PDF)

**S2 File. Complete search strings for all databases searched in this scoping review.**
(PDF)

**S3 File. List of papers excluded due to the inability to obtain full-text documents.**
(PDF)

## Acknowledgments

We thank Sandra Campbell from the University of Alberta Library for assistance in developing the search strategy. We also thank Dr Zvonimir Poljak, Dr Philipp Schott, Dr Okan Bulut, Giulia Scarpa, Nia King, and Carina de Micheli for their translating help within this project.

## Author Contributions

**Conceptualization:** Etienne J. de Jongh, Sherilee L. Harper, Shelby S. Yamamoto, Carlee J. Wright, Simon J. G. Otto.

**Data curation:** Etienne J. de Jongh, Soumyaditya Ghosh.

**Formal analysis:** Etienne J. de Jongh, Soumyaditya Ghosh.

**Funding acquisition:** Etienne J. de Jongh, Sherilee L. Harper, Shelby S. Yamamoto, Carlee J. Wright, Simon J. G. Otto.

**Investigation:** Etienne J. de Jongh.

**Methodology:** Etienne J. de Jongh, Sherilee L. Harper, Shelby S. Yamamoto, Carlee J. Wright, Craig W. Wilkinson, Simon J. G. Otto.

**Project administration:** Carlee J. Wright, Simon J. G. Otto.

**Resources:** Sherilee L. Harper, Simon J. G. Otto.

**Software:** Sherilee L. Harper.

**Supervision:** Sherilee L. Harper, Shelby S. Yamamoto, Craig W. Wilkinson, Simon J. G. Otto.

**Writing – original draft:** Etienne J. de Jongh, Simon J. G. Otto.

**Writing – review & editing:** Etienne J. de Jongh, Sherilee L. Harper, Shelby S. Yamamoto, Carlee J. Wright, Craig W. Wilkinson, Soumyaditya Ghosh, Simon J. G. Otto.

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
