## [Decision Letter · Decision Letter 0]

15 Jun 2021

PONE-D-20-34201

One Health, One Hive: A scoping review of honey bees, climate change, pollutants, and antimicrobial resistance

PLOS ONE

Dear Dr. Otto,

Thank you for submitting your manuscript to PLOS ONE. After careful consideration, we feel that it has merit but does not fully meet PLOS ONE’s publication criteria as it currently stands. Therefore, we invite you to submit a revised version of the manuscript that addresses the points raised during the review process.

We look forward to receiving your revised manuscript.

Kind regards,

Guy Smagghe, PhD

Academic Editor

PLOS ONE

Journal Requirements:

2. We note that Figure 3 in your submission contain map images which may be copyrighted. All PLOS content is published under the Creative Commons Attribution License (CC BY 4.0), which means that the manuscript, images, and Supporting Information files will be freely available online, and any third party is permitted to access, download, copy, distribute, and use these materials in any way, even commercially, with proper attribution. For these reasons, we cannot publish previously copyrighted maps or satellite images created using proprietary data, such as Google software (Google Maps, Street View, and Earth). For more information, see our copyright guidelines: http://journals.plos.org/plosone/s/licenses-and-copyright.

2.1.    You may seek permission from the original copyright holder of Figure 3 to publish the content specifically under the CC BY 4.0 license. 

2.2.    If you are unable to obtain permission from the original copyright holder to publish these figures under the CC BY 4.0 license or if the copyright holder’s requirements are incompatible with the CC BY 4.0 license, please either i) remove the figure or ii) supply a replacement figure that complies with the CC BY 4.0 license. Please check copyright information on all replacement figures and update the figure caption with source information. If applicable, please specify in the figure caption text when a figure is similar but not identical to the original image and is therefore for illustrative purposes only.

3. Please upload a copy of Supporting Information File S6 which you refer to in your text on page 37.

Reviewers' comments:

Reviewer's Responses to Questions

**Comments to the Author**

1. Is the manuscript technically sound, and do the data support the conclusions?

Reviewer #1: Yes

2. Has the statistical analysis been performed appropriately and rigorously? 

Reviewer #1: N/A

3. Have the authors made all data underlying the findings in their manuscript fully available?

Reviewer #1: Yes

4. Is the manuscript presented in an intelligible fashion and written in standard English?

Reviewer #1: Yes

5. Review Comments to the Author

Reviewer #1: This manuscript is about a scoping review to investigate interactions between climate change and AMR.

The authors describe that 37 articles were excluded because they were unable to locate full-text pdfs. I wonder which efforts were done to get access to these papers. The authors could be invited to send reprints or by interlibrarian service you could have asked for a copy of the paper. Are the authors sure that you didn't miss valuable information doing so?

I have checked a few references and I believe that the numbering is not correct. For instance page 20, lines 270-271, the paper on horizontal gene transfer is definitely 58 and not 59. The same on page 21, lines 282-285: this should be 61 and 62, and not 63. Please check the entire document carefully.

I would like to see some more criticism in the discussion section. For instance horizontal gene transfer: how frequently has it been described in the context of bee pathogens?

Page 21, lines 276-278: do you suggest here that cell membrane transporters are used to eliminate antimicrobials from the cytoplasm? Please explain better.

Another criticism that is lacking. In the paper of Gregorc et al (19) the authors examined gene expression of pesticide exposed in vitro reared larvae. Differences were found in some immunity-related genes like AMPs. However, challenge infection was done with varroa mites and not with bacteria or viruses. Does this allow to make any conclusions about immuno-competence? May be yes, but it is important to dig deeper in the refered papers.

6. PLOS authors have the option to publish the peer review history of their article (what does this mean?). If published, this will include your full peer review and any attached files.

Reviewer #1: No

---

## [Author Response · Author response to Decision Letter 0]

4 Aug 2021

Reviewer’s Comments to the Author

Reviewer #1: This manuscript is about a scoping review to investigate interactions between climate change and AMR.

Comment 1: The authors describe that 37 articles were excluded because they were unable to locate full-text pdfs. I wonder which efforts were done to get access to these papers. The authors could be invited to send reprints or by interlibrarian service you could have asked for a copy of the paper. Are the authors sure that you didn't miss valuable information doing so?

Response: Lines 153-158 (version with tracked changes)

Please see the added text to address the concern about missing articles. Second, in our review of this list, we were able to identify one additional article that was now available to us through our library. We have now included 22 articles in the study and revised the text, numbers, and citations throughout the document to reflect this. We recognize the importance of the remaining 36 excluded articles and ensured that every effort was made to locate them. We first utilized both University of Alberta and University of Guelph libraries, and when those resources were exhausted we turned to the interlibrary loan programs at both institutions to recover the remaining articles. Through the interlibrary loan we were able to locate a further 6 articles which we included within the screening process. The remaining 36 articles could not be found. We have described this extra step within our manuscript.

Comment 2: I have checked a few references and I believe that the numbering is not correct. For instance page 20, lines 270-271, the paper on horizontal gene transfer is definitely 58 and not 59. The same on page 21, lines 282-285: this should be 61 and 62, and not 63. Please check the entire document carefully.

Response Lines 478-639 (version with tracked changes): Thank you for finding this error. We have fixed and included the corrected bibliography, including the added reference mentioned above.

Comment 3: I would like to see some more criticism in the discussion section. For instance horizontal gene transfer: how frequently has it been described in the context of bee pathogens?

Responses:

Lines 368-377 (version with tracked changes): We appreciate the reviewer’s concern about being critical of the included literature. We did not conduct a risk of bias assessment as it is not a requirement for systematic scoping reviews according to JBI and PRISMA-ScR. However, we have included some statements in this section about the quality of the included literature and have highlighted some specific concerns.

Lines 290-293 (version with tracked changes): Regarding HGT, this was not a specific theme that emerged from the scoping review. We brought in the idea about HGT to link to the broader discussion on honey bee immunity and the idea insects could actually transfer genes back and forth between bacteria. As papers that linked AMR and CC/EP did not include HGT directly, we did not assess this frequency. We merely posit this as an area for future interest and research as it was not a direct finding of the review. As a result, we do not spend time formally evaluating the merits of the specific papers and conclude the paragraph by stating that evidence of transfer of AMR genes through these mechanisms remains largely unstudied. Further evaluation of these mechanisms are outside of the scope of this paper, which is focused on the link between AMR and Climate Change or AMR and environmental pollution in honey bee health. We brought in this idea from papers that supported the idea linked to changes in the microbiota of the honeybees and the fact that they are continuously exposed to pesticides and then transfer these things to pathogens in the honeybee gut.

Comment 4: Page 21, lines 276-278: do you suggest here that cell membrane transporters are used to eliminate antimicrobials from the cytoplasm? Please explain better.

Response Lines 290-298 (version with tracked changes): Thank you for highlighting this need for clarification. We have modified the text to better explain the causal pathway from transporter upregulation to AMR risk.

Comment 5: Another criticism that is lacking. In the paper of Gregorc et al (19) the authors examined gene expression of pesticide exposed in vitro reared larvae. Differences were found in some immunity-related genes like AMPs. However, challenge infection was done with varroa mites and not with bacteria or viruses. Does this allow to make any conclusions about immuno-competence? May be yes, but it is important to dig deeper in the refered papers.

Response Lines 245-252 (version with tracked changes): Thank you for this important comment. We recognize the need to clarify how we decided to “bin” articles. Although a full discussion of honey bee immune pathways is beyond the scope of this paper, we have made edits to better acknowledge how pest and parasite exposure is linked to immunocompetence. 

To provide further clarification, we explain the specific case of Gregorc et al. (2012) here for the reviewer’s benefit.

Morbidity as a result of Varroa mite exposure occurs via cellular invasion of Deformed Wing Virus, Escherichia coli, or other secondary infection. This clarification has been included.

Gregorc et al. (2012) explored immunocompetence via titres of deformed wing virus as well as other secondary pathogens in response to varroa mite exposure. “Loads for DWV and IAPV were elevated in bees challenged with Varroa (Fig. 1A), an expected result given that Varroa is a potential vector of these and other honey bee RNA viruses (Chen and Siede, 2007). This result was confirmed for Deformed Wing Virus in individual larvae, whereby larvae exposed to mites had a 900-fold higher average load for DWV (n = 113 and 94 assayed bees; Fig. 2).“

Table 2 line 19 in our manuscript highlights this viral focus of the paper and therefore its “immunocompetence tag”. The challenge by Varroa mite in this case was listed only under health aspects of concern, while the immunocompetence conclusions were drawn from the microbe of interest--deformed wing virus. 

Further, honey bee immune responses overlap significantly for various types of pathogens. This generalization is implied Gregorc et al. in their reference to “xenobiotic detoxification.” For example, antimicrobial peptides such as defensin have wide broad spectrum efficacy and are upregulated in response to Fungi, Bacteria, Viruses, and parasites (see DOI: 10.2478/v10289-012-0013-y).

---

## [Editor Report · Decision Letter 1]

25 Jan 2022

One Health, One Hive: A scoping review of honey bees, climate change, pollutants, and antimicrobial resistance

PONE-D-20-34201R1

Dear Dr. Otto,

We’re pleased to inform you that your manuscript has been judged scientifically suitable for publication and will be formally accepted for publication once it meets all outstanding technical requirements.

Kind regards,

Guy Smagghe, PhD

Academic Editor

PLOS ONE
---

## [Editor Report · Acceptance letter]

7 Feb 2022

PONE-D-20-34201R1 

One Health, One Hive: A scoping review of honey bees, climate change, pollutants, and antimicrobial resistance 

Dear Dr. Otto:

I'm pleased to inform you that your manuscript has been deemed suitable for publication in PLOS ONE. Congratulations! Your manuscript is now with our production department. 

Kind regards, 

on behalf of

Prof. Guy Smagghe 

Academic Editor

PLOS ONE